# Symbiotic Plant Biomass Decomposition in Fungus-Growing Termites

**DOI:** 10.3390/insects10040087

**Published:** 2019-03-28

**Authors:** Rafael R. da Costa, Haofu Hu, Hongjie Li, Michael Poulsen

**Affiliations:** 1Section for Ecology and Evolution, Department of Biology, University of Copenhagen, Universitetsparken 15, 2100 Copenhagen East, Denmark; rafael.dacosta@bio.ku.dk (R.R.d.C.); haofu.hu@bio.ku.dk (H.H.); 2Department of Bacteriology, University of Wisconsin–Madison, Madison, WI 53706, USA; hli555@wisc.edu

**Keywords:** carbohydrate-active enzymes, Blattodea, Macrotermitinae, microbiota, social insects, *Termitomyces*

## Abstract

Termites are among the most successful animal groups, accomplishing nutrient acquisition through long-term associations and enzyme provisioning from microbial symbionts. Fungus farming has evolved only once in a single termite sub-family: Macrotermitinae. This sub-family has become a dominant decomposer in the Old World; through enzymatic contributions from insects, fungi, and bacteria, managed in an intricate decomposition pathway, the termites obtain near-complete utilisation of essentially any plant substrate. Here we review recent insights into our understanding of the process of plant biomass decomposition in fungus-growing termites. To this end, we outline research avenues that we believe can help shed light on how evolution has shaped the optimisation of plant-biomass decomposition in this complex multipartite symbiosis.

## 1. Introduction

### 1.1. Plant Substrate Use as Main Nutrient Source

Plant biomass is the largest carbon reservoir on Earth and is used by a wide range of different organisms as a main food source [1]. A barrier in gaining nutrients from plant material is the inability of most animals to process plant biomass, due to the complexity of the plant cell wall [1], composed primarily of cellulose, hemicellulose, pectin and lignin [2]. The plant cell walls form a barrier to nutrient acquisition, and depending on the developmental stage [3], plant species [4], and degree of decomposition [5], structural components change in abundance. Breaking down this structural heterogeneity requires enzymatic, chemical, and/or mechanical reactions [5,6,7,8]. The enzymes for the breakdown, biosynthesis and modification of glycoconjugates, di-, oligo- and polysaccharides, are known as Carbohydrate-Active enZymes (CAZymes) [9]. No living organism has the complete metabolic reservoir necessary to convert plant cell wall components into nutrients [1]. To overcome this challenge, many organisms obtaining their nutrition from plant biomass engage in symbioses with diverse lignocellulolytic microorganisms [10,11,12,13,14].

### 1.2. Termites Have Relied on Symbiotic Digestion of Lignocellulose for Millions of Years

Among the insects, termites have achieved an outstanding ecological success, with more than 3000 extant species in 281 genera and eight families [15,16,17,18] that are widely distributed around the globe, including in tropical, subtropical and warm temperate regions. Termites evolved from a cockroach ancestor (Blattodea) [17] and are broadly divided into the lower (families: Mastotermitidae, Stolotermitidae Hodotermitidae, Archotermopsidae, Kalotermitidae, Serritermitidae and Rhinotermitidae) and higher (family: Termitidae) termites based on the respective presence or absence of intestinal flagellates [19,20].

Termites ingest lignocellulosic substrates at different degrees of decomposition [16,21] and have been classified in feeding groups based on their substrate use: Group I: lower termites feeding on wood, grass, and litter; Group II: higher termites feeding on wood, grass, and litter, including the fungus feeders (Macrotermitinae) in the sub-group IIF; Group III: highly-degraded wood and soil with a high organic content; and Group IV: the true soil-feeders [22]. Termites are among the few animals capable of producing endogenous cellulolytic enzymes [23,24], but this capacity alone is insufficient to decompose plant biomass [1]. Thus, they rely on their symbiotic gut microflora to depolymerize lignocellulose with subsequent fermentation, resulting in the production of short-chain fatty acids that can be oxidized by the host [25,26].

The wood-feeding lower termites associate with cellulolytic flagellates and gut bacteria, of which the most abundant are in the phyla Spirochaetes and Proteobacteria [25,27,28]. Approximately 60 million years ago (MYA), the ancestor of the higher termites lost the gut flagellates [29,30] and associated with an almost exclusively bacterial gut microbiota [31,32,33]. Spirochaetes, Fibrobacteres, and members of the TG3 phylum dominate higher termites feeding on sound wood or grass, while humus, soil, and fungus feeders have more similar gut communities, dominated by Firmicutes, Bacteroidetes and Proteobacteria [34]. However, they differ in the abundance of Spirochaetes, which is lower in soil feeders and almost absent in the fungus feeders [34].

Approximately 30 MYA, the higher termite subfamily Macrotermitinae engaged in an obligate co-dependent mutualism with basidiomycete fungi in the genus *Termitomyces* (Agaricomycetes, Lyophyllaceae) [35,36,37]. Fungus-growing termites comprise 11 genera with approximately 330 described species [33,35] that associate with ca. 40 described *Termitomyces* species [37,38]. In addition to the mutualism with *Termitomyces*, the termites maintain complex gut microbial communities [32,34,39,40,41,42]. The evolution of fungiculture involved the consequential origin of a dual decomposition strategy with complementary contributions to plant-biomass decomposition between the externally-maintained fungal gardens and bacterial contributions during two gut passages (Figure 1). This strategy appears to have allowed the subfamily to obtain near-complete decomposition of plant biomass, possibly contributing to their dominance as decomposers in the ecosystems they inhabit [5,43].

## 2. The Tripartite Fungus-Growing Termite Symbiosis

### 2.1. The Symbiosis Between Fungus-Growing Termites and Termitomyces

Prospective queens and kings are produced in mature nests. Then, during the mating flight, they leave their natal nests, pair-up, shed their wings and dig into the ground to establish a new colony [36]. Shortly thereafter, they begin to produce the first cohort of workers, who feed on soil and build pillars comprised of faecal pellets [44,45,46]. These first foragers also collect plant substrate, first turning these pillars greenish and a few days later they will be covered in *Termitomyces* hyphae [44,45,47]. Workers thus appear to obtain *Termitomyces* spores when foraging for plant substrates [48] and these spores are released into the environment from fruiting bodies (mushrooms) on mature nests [37,49]. This means that the transmission of *Termitomyces* is predominantly horizontal (environmental acquisition), but two exceptions to this pattern exist: the termite species *Macrotermes bellicosus* and the genus *Microtermes* transmit *Termitomyces* vertically (from parent to offspring colonies) [37,48,50].

The established fungus gardens appear as a cork-like structure termed the “fungus comb”. This comb is composed of termite primary faeces, which is a blend of plant material and asexual *Termitomyces* spores that pass through the guts of young workers [51] (Figure 1). The termites provide the fungal symbiont with optimal growth conditions (e.g., controlled temperature and humidity and inhibition of other fungi), and constant inoculation of plant substrate [52]. In return, *Termitomyces* decomposes plant material that cannot be digested by the termites themselves and provides nutrient-rich nodules formed by a conglomerate of conidiospores [47,51] (Figure 1).

Despite predominant horizontal transmission, phylogenetic analyses of Macrotermitinae and *Termitomyces* indicate some degree of interaction specificity, i.e., species of termites are restricted to associate with certain *Termitomyces* species [37]. At lower levels, specificity differences also remain; e.g., *Macrotermes natalensis* colonies associate with a single biological species of *Termitomyces,* whereas individual *Odontotermes* species may associate with several *Termitomyces* species [54,55,56,57]. Geographical isolation, synchronised dispersal of winged reproductives [56], and substrate use have been proposed to help explain these patterns [57,58,59]. Specificity in the light of horizontal transmission might appear counterintuitive, as vertical transmission often leads to a higher degree of interaction specificity and co-evolution [60]. However, it is often observed that traits of a symbiont are lost because their functions become redundant if the other partner reliably provides the resources [61,62]. This bilateral specialization between symbionts favours obligate associations, potentially leading to co-cladogenesis even in the absence of vertical transmission [56], as appears to be the case of the Macrotermitinae-*Termitomyces* association [37,51].

### 2.2. The Symbiosis between Fungus-Growing Termites and Gut Bacteria

The external primary decomposition of plant material in fungus gardens substantially changed the need for internal cellulose digestion, reflected in the marked difference between the gut microbiota of other higher termites and the fungus-growing termites, who host gut bacteria with reduced capacity for digestion of cellulose and other complex polysaccharides [39,63]. The possible roles of gut bacteria in fungus-growing termites include decomposition of other parts of the ingested plant substrates [32,39,63,64], inhibition of pathogens [65], amino acid synthesis [66], and nitrogen fixation [64,67]. The guts of fungus-growing termites also have a greater abundance of enzymes targeting chitin [39,64,68] and other fungal cell wall components [39,63,68], possibly in part contributed by the bacteria dominating fungus-growing termite guts [39,63,67].

Since fungus-growing termite species have many traits in common (e.g., plant substrate processing and a fungal diet), it is unsurprising that many bacterial taxa are shared across fungus-growing termite genera [42,69,70,71]. This ‘core’ microbiota is dominated by Bacteroidetes and Firmicutes and is more similar to cockroach gut communities than to those of most other termites [69,70,71,72]. Within colonies, gut microbial assemblies can also reflect specific termite colony member roles, with quantitative differences in bacterial relative abundances between castes and ages of workers and soldiers [41,42,73]. The most divergent microbiota is those in queens and kings, which are greatly reduced in bacterial diversity compared to workers and soldiers, being dominated by a few bacteria that are absent or only present in very low abundances in sterile (non-reproducing) castes [39,42,43,69]. This suggests that if male and/or female alates bring the bacterial inocula for the first workers in incipient nests, most of these are lost as the royal pair matures, likely due to changes in bacterial roles and the royal pair diet [42].

### 2.3. Substrate Use by Different Fungus-Growing Termite Species

Fungus-growing termites play important roles in recycling of nutrients in their environments. Macrotermitinae may harvest 20%–30% of the annual litter production and up to 65% of dry litter, while 80% of the carbon ingested by the Macrotermitinae may be digested by *Termitomyces* [74]. In some arid tropical environments, fungus-growing termites may recycle up to 90% of all dead plant material [75], benefitting natural ecosystems [76], but also causing serious damage and economical loss in agriculture [77,78,79,80,81,82].

Figure 2 provides an overview of known substrate use from studies on fungus-growing termite species in natural and in agricultural areas (for a full list, see Appendix A). Based on this, fungus-growing termites are best-characterised as generalists [18,83,84], with wood and grass being the most frequently used substrates. However, some termite species may preferentially forage on certain substrate types [85]. The size of plant biomass fragments [85] and seasonal variation [37] may also affect foraging preferences, which could also be driven by plant community composition affecting substrate availability. For example, fungus combs of *Macrotermes michaelseni* in Kenya were composed of 30% wood and 70% grass in one area and 64% wood and 36% herbaceous species in another area [84]. Such geographic variation suggests dietary flexibility, which may well contribute to termite abundance and their prominent role in nutrient recycling in African savannah ecosystems.

### 2.4. Plant Biomass Processing and Breakdown

The major components of plant cell walls, cellulose, hemicellulose, and lignin require mechanical, enzymatic or chemical reactions to break. Cellulose is a polymer of glucose linked with β-1-4 bonds and three types of enzymes are needed for its complete degradation: endo-cellulases cut the long cellulose chains into smaller chains, thereby forming ends that exo-cellulases can act on [24]. Exo-cellulases cleave to form disaccharides (cellobiose) from the longer cellulose chains, and cellobiases or glucosidases cleave this cellobiose into glucose, which can be taken up and utilised by the organism [24]. In contrast to recalcitrant cellulose, hemicelluloses are more easily hydrolysed either chemically or enzymatically and many enzymes (hemicellulases) contribute to doing so [24]. Lignin is a complex of phenolic rings, which are very difficult to cleave, leaving only white-rot fungi and some bacteria able to do so [24]. These organisms employ oxidizing enzymes (peroxidases and laccases) that create chain reactions, turning the aromatic rings into reactive free radicals [86]. Lignin does not contain nitrogen and the process likely also does not generate much energy, so cleaving lignin is likely mainly to improve access to the cellulose and any nitrogen within in the woody substrate [24,86].

Efficient plant biomass processing and decomposition in the fungus-growing termite symbiosis involve intricate steps across space (different locations within colonies) and time (different stages of biomass break down), including enzyme contributions from all partners in the symbiosis [39,63] (Figure 1). The enzymes involved in this breakdown have been the focus of many studies over the past decades; however, it remains unclear how generalisable these patterns are and how 30 million years of (co)evolutionary change has impacted patterns of symbiotic complementarity.

Differences in CAZyme profiles and expression across *Termitomyces* species are likely primarily driven by what is coded for in their genomes or what is required at a given point in time, i.e., dependent on what substrates the termites harvest. Johjima et al. [87] identified a wide range of CAZymes in *Macrotermes gilvus*-associated *Termitomyces* and found that most of these enzymes were pectin degrading, suggesting that foraging by *Macrotermes gilvus* on mainly fresh plant material influences the fungal symbiont enzyme potential and/or expression. Consistent with this assertion, da Costa et al. [5] found high expression of cellulases, laccases, and some hemicellulases in *Termitomyces* from *M. natalensis* and *Odontotermes* sp. foraging on dead plant material and animal dung [5].

Lignin breakdown in the symbiosis has also received attention without reaching a clear conclusion about how generalisable the process is across termite and fungal species. *Termitomyces* associated with *M. bellicosus* decomposes lignin to facilitate termite access to cellulose [87,126], but laccase activity, presumably contributing to lignin cleavage, has only been found in *Termitomyces* fungus combs associated with some (e.g., *Microtermes* sp., *Odontotermes* sp., and *Macrotermes gilvus*) but not other (*Odontotermes longignathus* and *Hypotermes* sp.) termite species [126]. An active laccase has further been proposed to be insufficient to cleave lignin, as the enzyme is unable to oxidise lignin by itself [87], but RNAseq from *M. natalensis* and *Odontotermes* spp. found that several enzymes targeting lignin can be present and expressed in at least *Termitomyces* species associated with these termite species [5]. The breakdown of lignin may also be complemented by chemical reactions during the first gut passage [53]. Given that specific cleavage and removal of lignin has been documented in wood feeders [127,128], it may not be surprising that the fungus-growing termite symbiosis depolymerizes lignin structures, even though only a few lignin-targeting bacterial enzymes have been identified in the termite gut [39,68].

The roles of gut bacteria in plant decomposition may vary with different termite-fungus-bacteria combinations, possibly in ways where the collective assembly of symbionts complements each other enzymatically. Liu et al. [64] identified xylanases and β-glucosidases from gut bacteria in *M. annandalei* and later complemented this with next-generation sequencing technologies on *O. yunnanensis* to identify a broad array of CAZyme genes [63]. These analyses suggested that a large portion of the bacteria-derived enzymes target oligosaccharides, which was corroborated in a comparison of *M. natalensis* and *O. yunnanensis* gut metagenomes with the dung-feeding termite *Amitermes wheeleri* and two *Nasutitermes* spp. (wood-feeders) [129,130]. The results indicate that enzymes targeting complex plant polysaccharides are relatively low in abundance in fungus-growing termite gut bacteria, while enzymes targeting oligosaccharides are relatively more abundant [39]. The enzymatic capacity of *Termitomyces* to degrade complex polysaccharides could thus be complemented by gut bacterial enzymes [39].

More recently, da Costa [5] compared the enzyme diversity and activity in nodules, worker guts, fresh and old comb in *M. natalensis* and two *Odontotermes* species and complemented this with RNAseq from nodules, fresh and old comb. A wide range of enzymes was identified, with the highest activity and expression being of cellulases and hemicellulases, and comparable nodule and worker gut enzyme activities suggest that enzymes within nodules remain active during gut passage [5]. After normalization of enzyme activities (i.e., enzyme activity/fungal biomass), old workers were most similar to old comb in their expression and young workers most similar to nodules and fresh comb, mirroring what differently-aged workers eat (Figure 1). Although enzyme activity was higher in nodules and fresh comb than old comb, fungus comb RNAseq suggested that the highest expression of these enzymes is in the old comb. This may imply that enzymes are produced in the mature older parts of the comb and transported to the nodules via *Termitomyces* hyphae, allowing for transfer through worker guts to the fresh comb, where the enzymes are needed to cleave components in the freshly-incorporated plant substrate (Figure 1). This supports the “ruminant hypothesis” by Nobre and Aanen [58], who hypothesised that *Termitomyces* could use the first gut passage to efficiently move lignocellulosic enzymes from mature to fresh parts of the fungus comb.

## 3. Research Avenues to Improve Our Understanding of the Evolution of Ancient Symbiotic Plant Biomass Decomposition

Fungus-growing termites manage an elaborate tripartite symbiosis that appears to have overcome major challenges for efficiently utilizing plant biomass. The termites process and provide their microbial symbionts with substrate, and these microbial partners offer the genetic machinery necessary for complete utilisation of plant substrates. While recent years have provided many novel insights, our understanding of how evolution has shaped the optimisation of plant-biomass decomposition is still lacking in many aspects. Albeit not an exhaustive list, we believe that the set of research avenues we outline below will be important to improve our understanding of the fungus-growing termite symbiosis specifically and complex symbioses more broadly.

### 3.1. How has 30 Million Years of Evolution Altered Symbiotic Contributions to Plant-Biomass Decomposition?

We lack a fundamental understanding of differences in plant-biomass decomposition potential across different *Termitomyces* species, and how such differences may be complemented by different contributions from gut bacteria. Currently, we lack information from the vast majority of the ca. 40 described *Termitomyces* species, and comparative analyses of their genomes paired with metagenome studies on gut bacteria symbionts would allow for insights into the co-evolutionary patterns of CAZyme provisioning and division of symbiont labour in the symbiosis.

### 3.2. Improving Our Understanding of the Link between Enzyme Targets and the Producing Organisms

A major challenge in understanding functions within complex symbiont communities is assigning symbiont identities to roles. This is less problematic for the monoculture *Termitomyces* fungus maintained by the termites but challenging for the bacterial communities [34,41,42,69,70]. High-quality gut metagenomes would allow for better assemblies and binning of bacteria OTUs. This could be coupled with bioinformatic predictions of putative functions with e.g., Peptide Pattern Recognition [131,132], which uses binding-site identification from sequences to improve predicted enzyme functions. This could both help establish gut bacteria functions and clarify whether variation in gut bacteria community composition between termite species is relevant to plant biomass processing and division of labour between the termites, fungal symbiont, and gut bacteria.

### 3.3. How Variable Is Substrate Use across Termite Species

Our current understanding of substrate use is restricted to very broad categories (e.g., wood, grass, etc.) without the identification of plant species harvested (Figure 2). DNA metabarcoding of environmental DNA [133] could be employed on termite guts and fungus combs to establish what plant families, genera, and even species, termites forage on. This would allow us to establish whether generalist substrate use is the norm and differences merely reflect plant availability in the environment or if preferences indeed exist. Laboratory experiments providing the termites with various plant species or biomass at different degrees of decomposition could complement this to elaborate any termite preferences. Insights from such work could help inform how foraging affects processes on ecological (e.g., impacts on the environment) and evolutionary (e.g., how the adoption of new diets may shape ecological traits) time scales.

### 3.4. How Flexible Is Enzyme Production in Fungus-Growing Termite-Associated Symbionts?

Substrate preferences between termite species/genera could lead to specialisation in enzymatic machineries. Alternatively, enzyme production could be plastic depending on plant species availability and seasonality. These alternative hypotheses could be explored by coupling substrate preference determination with enzyme assays (e.g., chromogenic substrates, AZCL, lignin-degrading enzyme assays [5,134,135,136]) and symbiont CAZyme gene expression in fungus combs and termite guts after termite foraging on different substrates, either in natural environments over geographical locations with different plant communities or through laboratory experimentation.

### 3.5. How Do Caste Roles and Caste-Specific Symbionts Interact to Affect Decomposition?

Social insect castes based on individual age and size are important for colony function and the integration and decomposition of plant biomass. As elucidated above, gut microbial community compositions differ between fungus-farming termite castes [41,42,70], but the causal reasons for these differences are as of yet largely unclear. They may merely be driven by differences in diet between castes (workers eat plant substrate, while soldiers and reproductives do not), which could select for different bacteria to flourish within guts or lead to differences in bacterial contributions to the breakdown of dietary components. Alternatively, community differences may imply that different bacteria serve important functions that affect caste roles, such as aiding lignin cleavage in workers [53] or contributing to defensive compounds in soldiers. Work that can shed light on bacterial functions within gut communities thus has the potential to aid our understanding of the role of symbionts in a social evolution context.

### 3.6. Do Differences in Substrate Use Align with the Interaction Specificity between Termite Host and Symbionts?

The importance of substrate use for patterns of interaction specificity between the termites and *Termitomyces* could be tested by providing laboratory colonies with filter paper containing spores from multiple *Termitomyces* strains and/or even species. If the termites select their ‘normal’ symbiont in the presence of multiple symbionts, substrate type may not be the only factor of importance for interaction specificity. Providing the termites with different plant substrates containing spores of non-native symbionts could help establish whether termite species with high degrees of interaction specificity with *Termitomyces* could establish association with new symbionts, or whether adaptations and specificity preclude that such novel associations arise [44].

### 3.7. Does Fenton Chemistry Play a Role in Lignin Depolymerization?

Fenton chemistry (the Fenton reaction) is a process in which hydrogen peroxide (H_2_O_2_) in the presence of e.g., soluble iron is split to generate water and hydroxyl radicals (**^·^**OH—a reactive oxygen species that is a strong oxidizing agent). Non-enzymatic Fenton chemistry has been identified in other insect [137] and lower-termite [100] guts, where it has been proposed to play a role in gut-mediated lignocellulose breakdown. Several Auxiliary Activity (AA) families that could initiate Fenton reactions have been identified in *Termitomyces* RNAseq data [5] and in a *M. natalensis* gut metagenome [138], suggesting the potential for such reactions being of importance. Establishing whether this indeed is the case would be an exciting research avenue to further improve our understanding of the role of Fenton reactions in lignin depolymerization.

## 4. Conclusions

Recent opportunities in -omics approaches have provided substantial and novel insights to symbiont roles in plant biomass decomposition in fungus-growing termites. A number of pioneering studies have determined broad substrate use, suggesting that farming termites are generalist rather than host plant-specific. However, work that goes beyond characterisations of substrate use in these broad categories (wood, grass, etc.) could help shed light on cryptic specificities. Understanding substrate use would allow us to better evaluate the role of fungus-farming termites in nature, and to establish if substrate use plays a role in governing termite-symbiont association specificities.

The symbionts associated with the farming termites do not *per se* appear to differ substantially from other plant-biomass degrading microbes, suggesting that it is rather the integration of the external fungal comb and internal gut passages than novel enzymes for plant biomass decomposition that enable the symbiosis to digest plant polysaccharides. This makes the symbiosis interesting to compare to other plant biomass decomposition systems/symbioses (e.g., the cow rumen), which would allow us to establish how alternative strategies for efficient decomposition have been optimised by natural selection.

The patterns of specificity over the long evolutionary history of the association provide excellent opportunities for comparative analyses of substrate use and plant biomass decomposition. This also applies to the contributions of enzymes from fungal and bacterial symbionts, which currently suffer from being biased towards a few termite species, being focused on either bacterial communities or *Termitomyces* in isolation, and often overlooking termite enzyme contributions. A more holistic approach with comparative analyses of all partners in a (co)evolutionary context across phylogenies, geography, and habitats would improve our understanding of both individual symbiont assemblies and the evolutionary histories of conserved and derived plant-biomass decomposition strategies.

## Figures and Tables

**Figure 1 insects-10-00087-f001:**
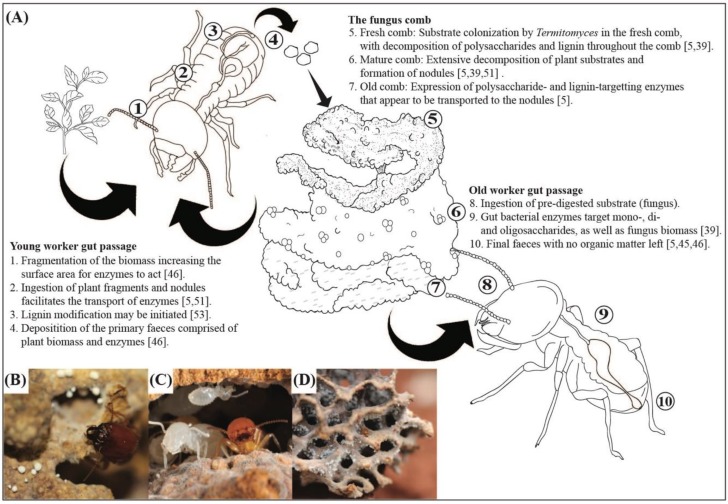
(**A**) The process of plant biomass incorporation and symbiotic complementary decomposition in the fungus-growing termites *Macrotermes* and *Odontotermes* species [5,46,51,53]. (**B**): *Macrotermes natalensis* soldier and nodules within the fungus comb (photo by M.P.). (**C**) *Macrotermes bellicosus* nymphs and workers in the fungus comb (photo by Nicky P.M. Bos). (**D**) *Odontotermes* sp. fungus comb with workers (photo by M.P.).

**Figure 2 insects-10-00087-f002:**
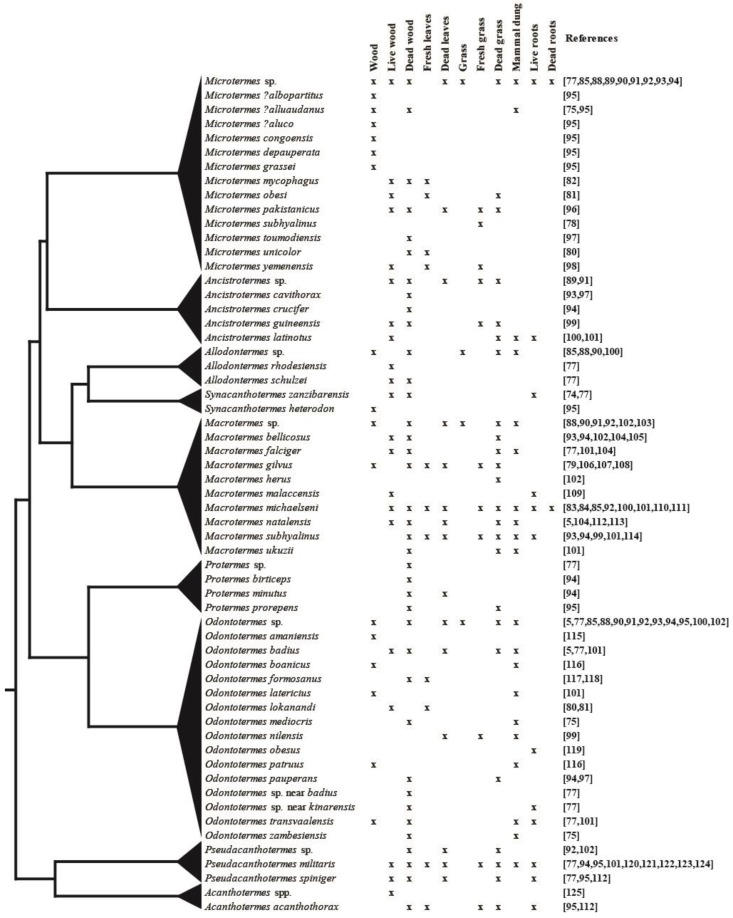
Forage substrate use by fungus-growing termites found in the literature (for a full list of all references and their reported findings, see Appendix A [5,74,75,77,78,79,80,81,82,83,84,85,88,89,90,91,92,93,94,95,96,97,98,99,100,101,102,103,104,105,106,107,108,109,110,111,112,113,114,115,116,117,118,119,120,121,122,123,124,125]) mapped on a schematic phylogeny of the subfamily [35]. Species given the same species name or labelled sp. or spp. in the original reports were grouped for clarity. The last columns “Wood” and “Grass” give cases where authors only mention forage substrate but not whether the plant material was alive or dead. The annotations “?” or “near” were not explained in the original reports, and the species were, therefore, treated as unique here.

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
