# Peer review of "Symbiotic Plant Biomass Decomposition in Fungus-Growing Termites"

_insects, 2019, doi:10.3390/insects10040087_

Round 1

Reviewer 1 Report

da Costa et al. have provided an interesting and well-researched review. The figures are clear and appropriate. The summary of feeding preferences across Macrotermitinae (Fig. 2) is a valuable service to the field. Overall the quality is high and I only have a few suggestions for improvement. 

There are several cases where more specifics would improve the review. For example, the description of colony establishment should be described in more detail in the text. The figure provides more information, but the figure should be a supplement, not the only source. Fig. 1 left me wondering whether “young worker” really meant a younger individual, or whether it meant worker of a young colony, among other things. This 10 step process of biomass breakdown should be explained in detail in the text, with appropriate references. This information would also help the reader to understand the final paragraph of section 2.4. Similarly, the last paragraph of section 2.1 states that Macrotermitinae and Termitomyces have “some degree of interaction specificity”. Later in the paragraph there is one sentence briefly mentioning the situation for Macrotermes and Odontotermes - please move this to immediately after the first sentence, and expand it to better explain the state of knowledge in more detail. In section 2.4, more background/specifics about the wood decomposition process, and lignin decomposition in particular, would be helpful. How well is it understood in other systems? What is the process and which enzymes are involved? How do Macrotermitinae differ in enzymes and process? The review summarizes what is known in Macrotermitinae, but not how that knowledge fits with the broader field. 

Additional suggestions below:

line 39 references are for phylogenetic analysis when the statement is about taxonomy - please include a reference to a taxonomic work such as Krishna et al. 2013

line 50 reference is for endogenous cellulases in Panesthia but the statement is about the distribution of cellulases across animals, so consider a review such as Cragg et al. 2015 Current Opinion in Chemical Biology

lines 82-84 this sentence is out of place. This information more properly belongs in the later paragraph about symbiont specificity and evolution

line 82, 98 and possibly others “horizontal transmission” is not the appropriate term; “environmental acquisition” would make more sense. Horizontal transmission implies transfer from one symbiotic system to another, in contrast to vertical transmission. 

line 103 insert “even” before “in the absence of”

line 127 replace “sterile” with “other” for clarity because sterile can also mean free of microbes

line 227 subheading is a statement but ends with a question mark - rephrase it as a question or else remove the question mark

line 242 “establish if” would be better as “establish whether”

line 249 remove “fixed or”

line 251 I think this sentence could end after “enzymatic machinery”. “upheld in the respective symbiosis to be optimally adapted to utilize given substrates” is overly wordy and, as far as I can tell, meaningless. 

line 259 remove “colony life and” 

line 259 the paragraph justifying question 3.5 is very short on specifics. What exactly are these caste-specific differences that have been found? Are they all in Macrotermitinae or are they in other groups? What might we expect to gain from understanding them better? 

line 275 (question 3.7) please provide an explanation of Fenton chemistry in the justification paragraph

line 278 remove hyperlink and replace with [99]

line 283 The conclusions section is a bunch of fluff. Try to provide some real information, for example by summarizing the current state of knowledge and the benefits of closing specific knowledge gaps. Avoid trite phrases such as “recent opportunities in -omics approaches” and “provide excellent opportunities … to help shed light on”

line 309 check that species names are italicized throughout this section

Author Response

da Costa et al. have provided an interesting and well-researched review. The figures are clear and appropriate. The summary of feeding preferences across Macrotermitinae (Fig. 2) is a valuable service to the field. Overall the quality is high and I only have a few suggestions for improvement. 

There are several cases where more specifics would improve the review. For example, the description of colony establishment should be described in more detail in the text. The figure provides more information, but the figure should be a supplement, not the only source. Fig. 1 left me wondering whether “young worker” really meant a younger individual, or whether it meant worker of a young colony, among other things. This 10 step process of biomass breakdown should be explained in detail in the text, with appropriate references. This information would also help the reader to understand the final paragraph of section 2.4.

Response: We had previously written a paragraph in the text that took readers through the process,

but decided that the figure would provide an easier way to follow this process. We still find that this is the case and have therefore not elaborated in detail on the process.

Similarly, the last paragraph of section 2.1 states that Macrotermitinae and Termitomyces have “some degree of interaction specificity”. Later in the paragraph there is one sentence briefly mentioning the situation for Macrotermes and Odontotermes - please move this to immediately after the first sentence and expand it to better explain the state of knowledge in more detail.

Response: We have followed these suggestions and revised accordingly (Lines 99-111).

In section 2.4, more background/specifics about the wood decomposition process, and lignin decomposition in particular, would be helpful. How well is it understood in other systems? What is the process and which enzymes are involved? How do Macrotermitinae differ in enzymes and process? The review summarizes what is known in Macrotermitinae, but not how that knowledge fits with the broader field. 

Response: We now provide a brief introductory paragraph to section 2.4 to set the stage for what is generally needed for the breakdown of plant cell wall components. The symbionts in the farming termite symbiosis do not per se differ from your typical plant-biomass degrading fungi/bacteria, which we now address in the revised Conclusion address this where we also highlight why there is a need for more work to elaborate how the termite symbiosis differs from other plant-biomass decomposition symbioses/environments (Lines 152-165).

Additional suggestions below:

line 39 references are for phylogenetic analysis when the statement is about taxonomy - please include a reference to a taxonomic work such as Krishna et al. 2013

Response: This is a good idea and we have added a reference to Krishna et al. (Line 40).

line 50 reference is for endogenous cellulases in Panesthia but the statement is about the distribution of cellulases across animals, so consider a review such as Cragg et al. 2015 Current Opinion in Chemical Biology

Response: We thank the reviewer for this suggestion and now also refer to Cragg et al. here (Line 51).

lines 82-84 this sentence is out of place. This information more properly belongs in the later paragraph about symbiont specificity and evolution

Response: We believe that by elaborating on the different transmission modes (see below) in this paragraph, the logic of having this sentence here is clearer. We have therefore not moved the sentence.

line 82, 98 and possibly others “horizontal transmission” is not the appropriate term; “environmental acquisition” would make more sense. Horizontal transmission implies transfer from one symbiotic system to another, in contrast to vertical transmission. 

Response: Vertical transmission is from parent to offspring (parent to offspring colony in social insects), while “the transmission of infections between members of the same species that are not in a parent-child relationship” is the definition of horizontal transmission. We now clearly explain what we mean with horizontal and vertical transmission in the context of the association with Termitomyces (Line 102-110).

line 103 insert “even” before “in the absence of”

Response: Has been revised as suggested.

line 127 replace “sterile” with “other” for clarity because sterile can also mean free of microbes

Response: Using the term sterile is common in social insects but to keep it clear that we refer to non-reproducing colony members, we inserted (non-reproducing) after sterile.

line 227 subheading is a statement but ends with a question mark - rephrase it as a question or else remove the question mark

Response: Has been revised as suggested.

line 242 “establish if” would be better as “establish whether”

Response: Has been revised as suggested.

line 249 remove “fixed or”

Response: Has been revised as suggested.

line 251 I think this sentence could end after “enzymatic machinery”. “upheld in the respective symbiosis to be optimally adapted to utilize given substrates” is overly wordy and, as far as I can tell, meaningless. 

Response: We agree and have revised as suggested.

line 259 remove “colony life and” 

Response: Has been revised as suggested.

line 259 the paragraph justifying question 3.5 is very short on specifics. What exactly are these caste-specific differences that have been found? Are they all in Macrotermitinae or are they in other groups? What might we expect to gain from understanding them better? 

Response: We have rewritten this section to better reflect what we believe will be of importance to improve our understanding of gut microbial differences between castes. We focus these suggested avenues to be on the Macrotermitinae, and therefore refrain from discussing other termites – although naturally similar work could be done in other sub-families as well (Lines 276-286).

line 275 (question 3.7) please provide an explanation of Fenton chemistry in the justification paragraph

Response: We now do so (Lines 297-299).

line 278 remove hyperlink and replace with [99]

Response: Has been revised as suggested.

line 283 The conclusions section is a bunch of fluff. Try to provide some real information, for example by summarizing the current state of knowledge and the benefits of closing specific knowledge gaps. Avoid trite phrases such as “recent opportunities in -omics approaches” and “provide excellent opportunities … to help shed light on”

Response: We agree and have rewritten the conclusion section (Lines 307-329).

line 309 check that species names are italicized throughout this section

Response: We have checked the reference list for both italics and other mistakes.

Reviewer 2 Report

This is the most recent in a flurry of reviews on the subject of the termite-fungus symbiosis in the subfamily Macrotermitinae, informed by genomic data. As such, it must be accepted as the latest word on the subject, although the extent to which it can be regarded as "termite ecology" is arguable, but this is a matter for the editors rather than the referees to decide. Also, it possibly relies rather heavily on the study of Macrotermes natalensis, and it's not entirely clear how much of the information furnished is from field collected specimens as opposed to laboratory cultures. There is also a surfeit of references to papers in press. 

line 12 and elsewhere. Is "herbivorous/herbivore" correct terminology, rather than "detritivore", the latter obviously applicable to termites by observation in the field except where dead or senescent plant detritus is scarce (for example after burning or weeding). Herbivory, sensu stricto, is confined to a few insect orders (Coleoptera, Diptera, Hemiptera, and even in these taxa is by no means universal) where it is accompanied by a suite of adaptations not reported in termites, fresh plant tissue being for the most part poisonous to all animals including humans (a few families containing our domesticated crop plants are the exceptions). Were termites not to have evolved a variety of symbioses capable of degrading lignocellulose, and at a later stage polyaromatic soil organic matter, their evolution and current dominant role in tropical soil ecosystem functions would not have emerged.

However, on the positive side the review does address the question of whether the Macrotermitinae are generalist rather than host plant-specific feeders, a matter on which earlier literature is unclear.

The very large number of references in the bibliography (137) makes the chapter especially useful, although quite a lot of pioneering work in the 1980s and 1990s is omitted and the conclusions referred to attributed to later authors, but this is simply a matter of damaged pride for the older generation of termite researchers and not a flaw in the current dialogue about fungus-growing termites.

Overall, an excellent article.

Author Response

This is the most recent in a flurry of reviews on the subject of the termite-fungus symbiosis in the subfamily Macrotermitinae, informed by genomic data. As such, it must be accepted as the latest word on the subject, although the extent to which it can be regarded as "termite ecology" is arguable, but this is a matter for the editors rather than the referees to decide. Also, it possibly relies rather heavily on the study of Macrotermes natalensis, and it's not entirely clear how much of the information furnished is from field collected specimens as opposed to laboratory cultures. There is also a surfeit of references to papers in press. 

Response: Fungus-growing termite research is undoubtedly biased by what species have been explored, and one goal of this review was to encourage comparative explorations across genera and species to expand on generalizability/variation in degradation processes (cf. Lines 322-329). We do not feel that we provide excessive citations of in press papers, as the two manuscripts in press are published in their final versions with open access on the publishers’ websites and the only ‘submitted manuscript’ is available from biorxiv.

line 12 and elsewhere. Is "herbivorous/herbivore" correct terminology, rather than "detritivore", the latter obviously applicable to termites by observation in the field except where dead or senescent plant detritus is scarce (for example after burning or weeding). Herbivory, sensu stricto, is confined to a few insect orders (Coleoptera, Diptera, Hemiptera, and even in these taxa is by no means universal) where it is accompanied by a suite of adaptations not reported in termites, fresh plant tissue being for the most part poisonous to all animals including humans (a few families containing our domesticated crop plants are the exceptions). Were termites not to have evolved a variety of symbioses capable of degrading lignocellulose, and at a later stage polyaromatic soil organic matter, their evolution and current dominant role in tropical soil ecosystem functions would not have emerged.

Response: We completely agree that termites should not be called herbivores and now refrain from doing so.

However, on the positive side the review does address the question of whether the Macrotermitinae are generalist rather than host plant-specific feeders, a matter on which earlier literature is unclear.

The very large number of references in the bibliography (137) makes the chapter especially useful, although quite a lot of pioneering work in the 1980s and 1990s is omitted and the conclusions referred to attributed to later authors, but this is simply a matter of damaged pride for the older generation of termite researchers and not a flaw in the current dialogue about fungus-growing termites.

Response: It was admittedly a task to try to gather these references, and we also should acknowledge that we do not mean to imply that we have found all. Our omissions are mainly due to the focus of the review, and recent papers are highlighted to provide an overview of our understanding of the symbiosis based on the most recent insights. We invite suggestions for additional work that the reviewer thinks we have wrongfully overlooked.

Overall, an excellent article.

Thank you for the nice comments. We appreciate the comment

Reviewer 3 Report

Line 36 - change they to  'many herbivores'

lines 82-84 - it would be helpful to briefly explain the difference between horizontal and vertical as it relates to fungal spore transmission.

Line 476 - Could not find reference 64 in text or tables.

Line 530 - Could not find reference in text or tables.

Author Response

Line 36 - change they to 'many herbivores'

Response: We now avoid using herbivores and have revised according to that change.

lines 82-84 - it would be helpful to briefly explain the difference between horizontal and vertical as it relates to fungal spore transmission.

Response: This is a good suggestion and we now do so.

Line 476 - Could not find reference 64 in text or tables.

Response: The reference has been removed from the reference list.

Line 530 - Could not find reference in text or tables.

Response: The reference has been removed from the reference list.